# *Lupinus albus* γ-Conglutin: New Findings about Its Action at the Intestinal Barrier and a Critical Analysis of the State of the Art on Its Postprandial Glycaemic Regulating Activity

**DOI:** 10.3390/nu14173666

**Published:** 2022-09-05

**Authors:** Giuditta C. Heinzl, Marco Tretola, Stefano De Benedetti, Paolo Silacci, Alessio Scarafoni

**Affiliations:** 1Department of Food, Environmental and Nutritional Sciences, Università degli Studi di Milano, 20133 Milano, Italy; 2Department of Veterinary Medicine and Animal Sciences, Università degli Studi di Milano, 26900 Lodi, Italy; 3Institute for Livestock Sciences, Agroscope, 1725 Posieux, Switzerland

**Keywords:** seed storage proteins, intestinal mucosa, glucose uptake, glycaemia, bioactive peptides, intestinal protein transport, α-amylase, α-glucosidase

## Abstract

γ-Conglutin (γ-C) is the glycoprotein from the edible seed *L. albus*, studied for long time for its postprandial glycaemic regulating action. It still lacks clear information on what could happen at the meeting point between the protein and the organism: the intestinal barrier. We compared an in vitro system involving Caco-2 and IPEC-J2 cells with an ex vivo system using pig ileum and jejunum segments to study γ-C transport from the apical to the basolateral compartment, and its effects on the D-glucose uptake and glucose transporters protein expression. Finally, we studied its potential in modulating glucose metabolism by assessing the possible inhibition of α-amylase and α-glucosidase. RP-HPLC analyses showed that γ-C may be transported to the basolateral side in the in vitro system but not in the pig intestines. γ-C was also able to promote a decrease in glucose uptake in both cells and jejunum independently from the expression of the SGLT1 and GLUT2 transporters.

## 1. Introduction

Legume seeds are known to bring various benefits to the diet thanks to their nutritional properties [1,2,3] and positive effects on human health [4,5,6]. Over the last two decades, lupins (*Lupinus albus*, *L. angustifolius*, and *L. mutabilis*) have attracted particular attention for their hypoglycaemic, hypocholesterolemic, and anti-inflammatory properties, especially regarding protein fractions [7,8,9,10]. Among them, γ-conglutin (γ-C) has been extensively studied for its postprandial glycaemic regulating activity in vitro [11,12], in vivo [13,14,15,16,17,18,19,20], ex vivo [21], and in humans [11,22,23,24]. The consumption of foods containing bioactive proteins and/or peptides that can help in managing different chronic diseases (e.g., diabetes) is nowadays gaining attention. The glycaemic regulating mechanism of action of γ-C has not been elucidated yet and the evidence for this bioactivity is still limited. Few studies have addressed their investigations towards the peptides obtained with gastrointestinal digestion [25]. The molecular mechanisms by which bioactive peptides are physiologically generated and likely reach and interact with the intestinal barrier where they can exert their effects still have to be studied and deepened. Hence, it is necessary to gather the information that leads to the clarification of these still-open questions, information that might be relevant for application in human health and nutrition.

This work focuses on the interactions of γ-C with an intestinal barrier, the contact point of a dietary protein with the organism, and its influences on glucose uptake. Finally, we deepen on γ-C potential in modulating glucose metabolism by assessing the possible inhibition of α-amylase and α-glucosidase.

## 2. Materials and Methods

### 2.1. Preparation of γ-Conglutin Hydrolysate

Purified γ-C protein was hydrolysed using a two-stage digestion with pepsin and pancreatin (cat. P-7012 and P-7545, respectively, Sigma-Aldrich, Saint Louis, MO, USA) (protein/enzyme ratio 30:1) to mimic gastrointestinal conditions. γ-C protein was hydrolysed with pepsin in HCl 1 mM pH 3.0 for 45 min and subsequently hydrolysed also with pancreatin in ammonium bicarbonate pH 7.5 for 45 min. Digestion was stopped by heating samples for 30 min at 65 °C. Inhibitory activity of the digested protein towards α-amylase and α-glucosidase was evaluated with two different types of digestion products: either by using pepsin and pancreatin, as described, or by using only trypsin at different digestion times (2, 15, and 45 min) to evaluate the changes that occur during the process.

A RP-HPLC method for the evaluation of γ-conglutin digestion was performed in a SIMMETRY300 C18 (5 µm) (4.6 mm × 250 mm) column (Waters, Sesto San Giovanni, Italy) fitted on a chromatographic apparatus (Waters) composed of two 510 HPLC Pumps, a 717plus Autosampler, and a 996 Photodiode Array Detector. Mobile phase flux was 0.8 mL/min, mixing solutions A (TFA 0.1% in water) and B (TFA 0.1% in ACN) as follows: 2 min isocratic 100% solution A, 50 min linear gradient to 25% solution A, and 75% solution B. Peaks were detected at 220 and 280 nm. SDS-PAGE was also performed according to Laemmli (1970) [26] on 15% polyacrylamide gels. The relative molecular mass of the polypeptides was determined by comparison with standard protein solution.

### 2.2. α-Glucosidase and α-Amylase Inhibition Assay

α-Glucosidase solution was prepared from intestinal acetone powders from rat (Merck Life Science, Milano, Italy). Briefly, 200 mg of rat intestinal acetone powder was dissolved in 4 mL of 50 mM ice-cold phosphate buffer (pH 6.8) and sonicated for 15 min at 4 °C. The suspension was vortexed for 20 min and then centrifuged at 10,000× *g* for 30 min at 4 °C. The resulting supernatant was used for the assay. The α-glucosidase inhibition assay was performed as follows: 20 μL of the enzyme solution, 50 or 100 μL of digested proteins (1.00 mg/mL), and 50 mM phosphate buffer pH 6.8 were mixed in 96-well plates to a final volume of 200 µL and incubated for 10 min at 37 °C. Then, 30 μL of 1 mM p-nitrophenyl-α-glucopyranoside (pNPG, Merck Life Science, Milano, Italy) was added as substrate and the mixture was further incubated at 37 °C for 1 h.

For pancreatic α-amylase assay, 10 μL of the enzyme solution (10 μM in CaCl_2_ 3 mM + NaCl 1 M), 50 or 100 μL of digested proteins (1.00 mg/mL), and 50 mM phosphate buffer pH 6.8 were mixed in 96-well plate to a final volume of 200 µL and preincubated for 10 min at 37 °C. Then, 15 μL of ceralpha (Megazyme, Bray, Wicklow, Ireland) was added to each well as the substrate and the mixture was further incubated at 37 °C for 5 min.

A positive control with acarbose (50 mM) and a sample blank with digested sample without γ-C were used as control. Sodium carbonate (100 mM) was added to stop the reaction. Both enzymes’ activities were determined by measuring the absorbance at 405 nm using an iMark^TM^ Microplate Reader (Bio-Rad, Hercules, CA, USA). Residual enzyme activities were calculated as follows:Activity % = ((Sample or inhibitor Abs − blank Abs)/Control absorbance) × 100

### 2.3. Cells Culture Conditions

Human intestinal epithelial Caco-2 cells were cultured in DMEM (Merck Life Science, Milano, Italy), supplemented with heat-inactivated foetal bovine serum (10%), 5% antibiotic solution, 5% L-glutamine (Merck Life Science, Milano, Italy). The cells were grown under 5% CO_2_ in a humidified air atmosphere at 37 °C and given a fresh medium every 2 or 3 days. For transport studies, Caco-2 cells were cultured (1 × 10^5^ cells/mL) on permeable polycarbonate filter supports in a 6-well plate using cell culture inserts with 0.4 µm pore size (cat. 353090, Falcon-Corning, Tewksbury, MA, USA) and grown for 21 days. After complete differentiation, cells were washed with PBS and supplemented with 2 mL of serum-free DMEM containing different concentrations of γ-C (0.50 and 1.00 mg/mL) both intact and digested in the apical chamber and 2 mL of PBS supplemented with Ca^++^ and Mg^++^ in the basolateral chamber. Control cells were supplemented by adding the same volume of DMEM and PBS. Apical and basolateral samples were collected after 4 h of incubation at 37 °C under 5% CO_2_ and a cell viability assay was performed using the MTT (3-(4,5-dimethylthiazol-2-yl)-2,5-diphenyl tetrazolium bromide) assay developed by Mosmann (1983) [27]. To the cells, 50 µL MTT (5 mg/mL of media) was added and incubated for 4 h at 37 °C. After incubation, 500 µL of dimethyl sulphoxide was added in each well to solubilise the formazan crystals. Optical density was determined with iMark^TM^ Microplate Reader (Bio-Rad, Hercules, CA, USA) using an absorption spectrum at 570 nm. Cell viability of treated cells was calculated by comparing viability of treated cells with viability of control cells (100% viable cells).

IPEC-J2 cells were maintained in Dulbecco’s modified Eagle’s medium (DMEM)/Ham’s F12 supplemented with 10% porcine serum, 50 U/mL penicillin, and 50 U/mL streptomycin. For Ussing chamber experiments, cells were seeded in 12-mm inside-diameter polycarbonate-filter inserts with pores of 0.4 μm-diameter (Corning, Tewksbury, MA, USA) and 200,000 cells/cm^2^-density. The basolateral and apical compartments were filled with 1.50 and 0.50 mL of culture medium, respectively. Cells were cultured in these conditions for 10 d after confluence was reached. Upon differentiation, the porcine serum concentration was reduced to 1%. After 24 h of serum reduction, transepithelial resistance (TEER) was measured with an EVOM2 epithelial voltohmmeter (World Precision Instruments, Sarasota, FL, USA). As TEER values exceeding 200 Ω × cm^2^ are generally considered acceptable, only monolayers with a TEER above 200 Ω × cm^2^ were used for absorption experiments. For protein expression quantification, cells were seeded on 12-well plates at 300,000 cells/cm^2^ density in similar conditions as described above. Once differentiated, the cells were incubated for 24 h with two different concentrations of lyophilised predigested γ-C (0.25 and 0.50 mg/mL) compared with the untreated group used as negative control.

### 2.4. SGLT1 and GLUT2 Glucose Transporters Gene Expression

SGLT1 and GLUT2 gene expression analyses were carried out on Caco-2 cells after treatments with intact protein or peptides to evaluate the changes in gene expression of glucose transporters. Differentiated Caco-2 cells were seeded on Transwell inserts as previously described and, after 21 days, were incubated for 4 h treatment with 1.0 mg/mL of intact protein and peptides. The undifferentiated cells were seeded in a 24-well plate at 1 × 10^5^ cells/mL and incubated for 48 h. After confluence was reached, 4 h of incubation with intact protein and peptides at 1.0 mg/mL. RNA was then isolated using an Aurum^TM^ Total RNA Mini Kit (Bio-Rad, Hercules, CA, USA). A reverse transcription was carried out with iScript^TM^ Reverse Transcription Supermix for RT-qPCR (Bio-Rad, Hercules, CA, USA). A q-PCR assay was performed using SsoAdvanced^TM^ UniversalSYBR^®^ Green Supermix (Bio-Rad) in a CFX Connect Real-Time System (Bio-Rad, Hercules, CA, USA) and primers specific for human genes SGLT1, GLUT2 and GAPDH [28]. Thermocycling conditions were one cycle at 96 °C for 10 min and then 40 cycles at 95 °C for 15 s followed by 60 °C for one min before a dissociation stage.

Immediately after the experiment, two segments of ileum and jejunum were cut from the intestines and frozen. Total RNA samples were isolated from frozen intestinal fragments according to Aurum^TM^ Total RNA Mini Kit (Bio-Rad, Hercules, CA, USA) manufacturer instructions. A reverse transcription was carried out with iScript^TM^ Reverse Transcription Supermix for RT-qPCR (Bio-Rad, Hercules, CA, USA). A q-PCR assay was performed using SsoAdvanced^TM^ UniversalSYBR^®^ Green Supermix (Bio-Rad) in a CFX Connect Real-Time System (Bio-Rad, Hercules, CA, USA) and primers specific for pig genes SGLT1, GLUT2, and RPL4 [29]. Thermocycling conditions were one cycle at 96 °C for 10 min and then 40 cycles at 95 °C for 15 s followed by 68 °C for one min before a dissociation stage. The mRNA expression level relative to the control group of each gene was calculated using the 2^−ΔΔCt^ method in reference to GAPDH for Caco-2 cells and RPL4 for pig intestines.

### 2.5. Western Blots

After treatment, the cells were lysed in CelLytic M (Merck Life Science, Milano, Italy) complemented with protease inhibitors. After centrifugation at 12,000× *g* at 4 °C for 10 min, the protein concentration in the supernatant was determined using an Asys UVM 340 microplate reader (Biochrom Ltd., Cambridge, UK). Twenty micrograms of total protein extracts was denatured at 95 °C for 5 min, separated via 7% SDS-PAGE gel (polyacrylamide/bisacrylamide(37.5:1); 0.35 M Tris HCl, pH 8.8; SDS, 0.1%; ammonium persulfate, 0.05%; and tetramethylethylenediamine, 0.01%). Proteins were then blotted on a WesternBright polyvinylidene difluoride membrane (Witec AG, Sursee, Switzerland) at 90 V for 90 min using the Bio-Rad Trans-Blot Turbo Transfer System (Bio-Rad). The membranes were blocked for 60 min at room temperature with 5% bovine serum albumin. The following primary antibodies were used to incubate the membranes at 4 °C overnight: SGLT1 (Ab14686, Abcam plc, Cambridge, UK) diluted 1:500 in 0.10% Tween, in 1X PBS containing 5% bovine serum albumin and vinculin (V4505, Sigma-Aldrich) diluted 1:3000. For SGLT1, the secondary antibody was goat-anti-rabbit (A9169, Sigma-Aldrich) diluted 1:1000 in 0.1% Tween 20, in PBS containing 5% milk powder, whereas goat-anti-mouse (DC02L, Merck Life Science, Milano, Italy) diluted 1:3000 in the same buffer was used as the secondary antibody for the vinculin. Chemiluminescence signals were detected using a Quantum-Kit (Witec AG, Sursee, Switzerland). Western blots were quantified by measuring the intensity of the correctly sized bands using a Syngene G:BOX (Syngene, Cambridge, UK). The ratio of the intensities of the protein bands of interest vs. the housekeeping protein was calculated for each filter, and the ratios from different Western blot filters were used to determine protein abundance.

### 2.6. Ussing Chamber Experiments—IPEC-J2

To evaluate D-glucose active uptake across intestinal epithelial cells, previously treated IPEC-J2 monolayers (exposed area of 0.33 cm^2^) were mounted on an Ussing chamber. The chambers were filled with 4 mL of Krebs–Ringer buffer (115 mM NaCl, 2.4 mM K_2_HPO_4_, 0.4 mM KH_2_PO_4_, 1.2 mM CaCl_2_, 1.2 mM MgCl_2_, and 25 mM NaHCO_3_^−^. The serosal buffer (pH 7.4) also contained 10 mmol/l glucose as an energy source that was osmotically balanced with 10 mM mannitol in the mucosal buffer (pH 7.4). Buffers were continuously perfused with a 95% O_2_ and 5% CO_2_ gas mixture. The temperature was maintained at 37 °C by a circulating water bath. Using a computer-controlled device, the transepithelial potential difference (TEER) and short-circuit current (Isc) were continuously monitored. Cells’ monolayers were voltage clamped at 0 mV by an external current after correction for solution resistance. Under these conditions, the absorption of cations and the secretion of anions increased Isc. Cells were equilibrated for 20 min before the mucosal addition of 5 mM D-Glucose. The corresponding equimolar concentration of D-Mannitol was added into the serosal compartment. At the end of the experiment, forskolin (10 μM) was added to the serosal compartment to test cell viability.

### 2.7. Ussing Chamber Experiments—Jejunum and Ileum

Six female Swiss Large White pigs housed at Agroscope (Institute for Livestock Sciences, Posieux, Switzerland) animal facilities were used. The animals were slaughtered at 171 (sd 2.8) d of age at the research station abattoir after being fasted for approximately 15 h, according to the Swiss Cantonal Committee procedures. For each pig, two jejunal samples starting from the third meter distal to the pylorus and two Ileal samples immediately before the ileocecal valve were mounted in Ussing chambers (Physiologic Instruments, San Diego, CA, USA). Each segment was bathed on its mucosal and serosal surfaces (exposed area 1.0 cm^2^) with the same corresponding KRB buffer. Specifically, each chamber was filled with 5 mL KRB buffer consisting of (in mmol/L) the following: mucosal—113.6 NaCl, 5.4 KCl, 0.2 HCl, 1.2 MgCl_2_, 1.2 CaCl_2_, 21 NaHCO_3_, 1.5 Na_2_HPO_4_, 2.0 Mannitol, 20 HEPES (4-(2-hydroxyethyl)-1-piperazineethanesulfonic acid), 0.01 indomethacin; serosal—113.6 NaCl, 5.4 KCl, 0.2 HCl, 1.2 MgCl_2_, 1.2 CaCl_2_, 21 NaHCO_3_, 1.5 Na_2_HPO_4_, 2.0 Mannitol, 7.0 HEPES, 10 Glucose, 6.0 Na-gluconate, 0.01 indomethacin. After the stabilisation period, the predigested γ-C (1.00 mg/mL) was added to the mucosal side and aliquots of 500 μL were collected on both mucosal and serosal sides. After 30 min, aliquots from each chamber were collected again. Immediately after, 10 mM D-glucose was added to the mucosal buffer and equimolar mannitol was added to the serosal side. The total active transport through the tissue was verified by monitoring the change in Isc (ΔIsc). The TEER was also measured at 2 min intervals under current clamped conditions. At the end of the experiment, forskolin (10 μM) was added to the serosal compartment to test cell viability. A scheme of the experiment is reported in Figure 1.

### 2.8. Characterisation of Pig Intestinal Products

Aliquots of 500 μL collected on both mucosal and serosal sides from pig intestines were analysed through SDS-PAGE in reducing and nonreducing conditions. Silver staining was also performed according to Blum et al. (1987) [30] in order to identify the presence of low abundant peptides. After staining, the gels were scanned with VersaDoc imaging system Model 4000 (Bio-Rad, Hercules, CA, USA) and analysed with ImageJ software (version 1.53s, NIH, University of Wisconsin, Madison, WI, USA) [31]. A RP-HPLC method for detection and quantification of peptides was carried out [32] using the same condition reported above. The purified peak bands of ileum were digested and analysed through mass spectrometry.

### 2.9. Statistical Analysis

Data were analysed using IBM SPSS Statistics, version 24 (SPSS, Chicago, IL, USA) and presented as mean ± standard errors. The data were tested for normality with the Shapiro–Wilk test. For experiments with IPEC-J2 cell line, results were obtained on a minimum of a technical duplicate of four independent experiments. Results were tested with one-way ANOVA as the data were normally distributed. For Ussing chamber experiments using jejunum and ileum tissues, a mixed linear model was used, in which the pig was considered as a random effect. Differences between control and treatment groups were considered significant when *p* < 0.05. The built-in statistical methods of Prism 9.3.1 (GraphPad Software, Inc., San Diego, CA, USA) were used for α-amylase and α-glucosidase inhibition assay. Results were analysed with Dunnett’s multiple comparisons test.

## 3. Results

### 3.1. γ-Conglutin Hydrolysate Characterisation

γ-C was hydrolysed to mimic gastrointestinal conditions by using a combination of pepsin and pancreatin digestions [33]. The products were analysed by RP-HPLC and SDS-PAGE. Figure 2A shows the chromatograms of the digestion with trypsin only (DP), whereas Figure 2B refers to the double digestion with pepsin and pancreatin (DPP). In the first case, some intact protein is still observable; whereas, following the double digestion, no intact protein was detectable. The size of the DPP digested peptides was such that they could not be retained by the used 15% acrylamide SDS-PAGE gel, where no bands are visible (Figure 2C).

### 3.2. Effects of γ-Conglutin on D-Glucose Active Uptake and Glucose Transporters Protein Expression in IPEC-J2 and Pig Small Intestine

To evaluate D-Glucose active uptake across intestinal epithelial cells, previously treated IPEC-J2 monolayers were mounted on an Ussing chamber. Membrane epithelial integrity was pre-emptively checked. The permeability of the tight junctions was determined by measuring the transepithelial resistance (TEER) of cell monolayers. The incubation for 24 h with 0.25 and 0.50 mg/mL γ-C derived peptides did not affect the TEER of the IPEC-J2 cells monolayer compared with the control (*p* > 0.05), as reported in Figure 3.

Compared with the untreated cells, a reduction in the glucose-induced ΔIsc in the γ-C25 (*p* = 0.006) and the γ-C50 (*p* = 0.001) was observed. In addition, all groups positively responded to the forskolin added at the end of the experiment, confirming the viability of the cells. The results are reported in the following Table 1.

D-glucose uptake was also assessed in ex vivo experiments using pig jejunum and ileum segments. In detail, the active uptake was evaluated by comparing the simplified in vitro results with those obtained from intestinal tissue. Independently by the intestinal segments, γ-C decreased (*p* < 0.05) the D-Glucose-induced Δ-Isc compared with the untreated (UT) tissues. No significant effects were observed on the TEER of tissues compared with the UT group (Table 2).

After this last experiment, an aliquot of the medium of both apical and basolateral chambers was collected and analysed by RP-HPLC and SDS-PAGE. The analysed aliquots of ex vivo experiments showed that some hydrolytic activity occurs during the 30 min of the experiment, especially in the jejunum sample, and new peptides are generated (Appendix A). This suggests that the animal tissue still possesses some kind of enzymatic activity, which was confirmed by the study of Ozorio et al. [34], who described an endopeptidase activity in the intestinal brush border of piglets. So, the peptides undergo further digestion when they meet the epithelial barrier. The decrease in γ-C derived peptides can be also related to bonds formed with intestinal mucosa or if they were able to enter the enterocytes.

Transit of digested lupin protein through the pig intestinal segments did not occur, as shown in Figure 4. The chromatography did not evidence any relevant differences in the treated samples. However, a peak was observed at 36 min of elution in all basolateral compartments. This peak was collected and analysed by SDS-PAGE, which evidenced a polypeptide of around 65 kDa (Appendix A). Mass spectroscopy analysis indicated the polypeptide as pig albumin.

Finally, we assessed the modulation of the expression of SGLT1 and GLUT2 glucose transporters (Figure 5) in the two tested animal systems. Either jejunum or ileum segments showed a weak upregulation of GLUT2 (*p* < 0.05), whereas no fold changes were observed for SGLT1 (Figure 5A). Despite differences between the treatments in the glucose absorption obtained by the Ussing chamber, the 24 h incubation with γ-C25 and γ-C50 did not affect SGLT1 protein expression compared with the UT in the IPEC-J2 cell line (Figure 5B) (*p* > 0.05).

### 3.3. Protein Transport through Caco-2 Artificial Intestinal Barrier, Cell Viability, and Glucose Transporters Gene Expression

In order to translate the results obtained on animal systems, we performed some trials on human Caco-2 intestinal cells too. In this part of the study, we assessed first the viability of Caco-2 cells under treatment with the γ-C and its derived peptides by MTT assay. In line with previous evidence, the results confirmed that γ-C and its peptides exerted no cytotoxic effect (Appendix A).

Gene expression analyses were carried out on Caco-2 cells to evaluate the direct action of protein treatments on expression of SGLT1 and GLUT2 glucose transporters. Relative gene expressions of GLUT2 and SGLT1 in undifferentiated Caco-2 cells are shown in Figure 6. An increase in gene expression was observed for all tested proteins/peptides compared with the untreated controls. Intact protein was able to induce a double increase in gene expression compared with the peptides. On the contrary, the differentiated cells did not give rise to changes in protein expression (Appendix A).

Furthermore, a study was conducted to assess the transport of γ-C from apical to basolateral chambers of transwell inserts. In Figure 7, the chromatographic separations of the proteins in the apical and basolateral chambers are shown. On the apical chamber, proteins were applied at two different concentrations (0.50 and 1.00 mg/mL) compared with the control without any protein. The peak at 37 min is the γ-C and is present in both chambers, essentially confirming the ability of Caco-2 cells to transport the protein through a model of the intestinal barrier, as previously described by Capraro et al. [21]. Under the adopted experimental conditions, no new peptides were detectable in the basolateral chamber, likely indicating that no proteolysis of γ-C occurred during the transport. Alternatively, γ-C once internalised might be degraded inside the cells and the residual intermediate products could not be transported to the basolateral compartment [21,35].

### 3.4. Inhibition of α-Amilase and α-Glucosidase Activity

To study the potential modulating activity of γ-C derived peptides on glucose metabolism, we evaluated the inhibition of α-amylase and α-glucosidase enzymes. A positive control acarbose at 3 μM strongly inhibited both enzymes activity. The native γ-C protein (not hydrolysed) did not exhibit inhibitory action (Appendix A), whereas its hydrolysed form inhibited the enzyme regardless of the time of digestion. This indicated that the intact protein had to be digested by pepsin and trypsin to produce peptides able to inhibit the enzymes (Figure 8). To rule out the possibility that the observed inhibition could be the result of the used proteolytic enzymes, samples digested without conglutin were used as control.

## 4. Discussion

The goal of the present work was to study the biological effects of the peptides generated with simulated gastrointestinal digestion with ex vivo techniques, using pig intestines, and in vitro, with IPEC-J2 and Caco-2 cells. We investigated the variation in the expression of the intestinal glucose transporters GLUT2 and SGLT1 and the internalisation capacity of γ-C was evaluated through the use of Ussing chambers (UC) and transwell inserts. These and other intestinal segments from different animal species have been used to perform UC experiments. Given the similarities with the human gastrointestinal tract (GIT), pig intestinal segments are also commonly used for human studies, together with mice [36]. For studies involving animal nutrition or animal efficiency, pigs are again the most used [36].

Although the possible molecular mechanisms by which γ-C and its derived peptides exert the glycaemic regulating activity have been largely explored by using different kinds of cell model systems (intestinal, liver, pancreatic, myoblast, musculoskeletal cells) as summarised in the Introduction and Table 3, there are still many undefined aspects. Clear and conclusive information on what happens at the meeting point between the protein and the organism—namely, the intestinal barrier—is still lacking. Despite all the efforts, the mechanisms of action of this protein remain elusive and little progress has been made in elucidating its real contribution to observed bioactivity.

The experimental setting allowed us to investigate the γ-C transport from the apical to the basolateral compartments, its effects on the D-glucose active uptake and glucose transporters protein expression, and the effects on the transepithelial resistance (TEER). Both the in vitro and ex vivo methods used have limitations [37,38]. Caco-2 and IPEC-J2 cells allowed us to perform a “long-term” γ-C treatment, but missed the complex organisation typical of the intestine. Contrastingly, intestinal tissue segments used in the Ussing chamber still contain the morphological and physiological features of the intestine, but this method allows only a “short-term” γ-C treatment. This period could be too short to properly investigate the uptake dynamics of the γ-C and its impact on protein expression modulation. Nevertheless, the collected results provide new interesting insights into the local effects of γ-C. Several studies demonstrated that pig jejunum is the segment of the small intestine in which the most Na_2_^+^-dependent glucose transport occurs, while there are not many studies focused on the glucose uptake in the ileum, and the present study demonstrated that the pig ileal segment is strongly involved in the D-glucose uptake too. Those results are in line with another study performed on postweaning piglets [39]. γ-C was able to decrease the glucose uptake independently by the intestinal segment, demonstrating that the ileum is also a target of this molecule. This effect was not driven by a modulation in the SGLT1 protein expression, but a different status of the post-translational modification of the SGLT1 protein could be investigated in future studies. Finally, we studied its potential in modulating glucose metabolism by assessing the possible inhibition of α-amylase and α-glucosidase. RP-HPLC analyses showed that γ-C could be found in the basolateral compartment in the in vitro system but not in the pig intestines, where it causes tissue damage with albumin release. γ-C administration was able to promote a decrease in D-glucose uptake in both cells and jejunum independently from the expression of the SGLT1 and GLUT2 transporters and without detrimental effects on TEER, compared with the untreated counterparts. The enzyme’s inhibitory activity gave a slight but significant reduction. For this reason, in our study, we decided to compare cells in vitro with ex vivo pig intestine segments that represent the closest system to reality.

The collocation of the obtained results in the frame of the literature concerning γ-C produced so far is worth an effort. Thus, we decided to critically review the most recent and relevant works on the topic (Table 3). Regrettably, all these studies have been conducted on simplified cell systems. At the same time, it is worth pointing out that many bioactive peptides do not likely need to be absorbed in the gastrointestinal tract and may play a role via signalling pathways. With this consideration, however, for studying bioactivities exerted by a given protein assumed with the diet, the use of different kinds of cells apart from intestinal ones can appear inappropriate.nutrients-14-03666-t003_Table 3Table 3Conglutin-γ in vitro studies on different cell cultures.AuthorsYearCell CultureAnalysisSirtori et al. [40]2004HepG2Lipid-lowering mechanismTerruzzi et al. [12]2011C2C12P7056 kinase, eIF-4E, PHAS-1 activationCapraro et al. [21]2011Caco2Protein uptakeLovati et al. [14]2012HepG2glucose consumptionCapraro et al. [35]2013HepG2InternalisationLammi et al. [41]2016Caco2Intestinal absorptionWojciechowicz et al. [42]20183T3-L1Viability, proliferation, and adipogenic markersMuñoz et al. [43]2018Caco-2, 3T3-L1Glucose uptake, glucose transporter, PEPCK expressionBarbiroli et al. [44]2019Caco-2Peptide transporters expressionLima-Cabello et al. [45]2019PANC-1IRS-1, p85-PI3K, GLUT4 upregulation Guzman et al. [46]2021HepG2, Min6Cell death induced by lipotoxicity and oxidative stressTapadia et al. [25]2021INS-1E, BRIN-BD11, Primary human skeletal muscle myoblastsInsulin secretion, glucose uptake, glycogen content, protein synthesisGracio et al. [47]2021HepG2, rabbit erythrocytesGlycosylated receptors binding, hemagglutination assays


To lower the glycaemic index in diet, there is a growing body of research into products that slow the absorption of carbohydrates through the inhibition of enzymes responsible for their digestion [48]. Different enzymes have been tested such as α-amylase and glucosidase, DPP-IV inhibitors. Further, in this case, γ-C resulted to show a minimal inhibitory activity for α-glucosidase and a dose-dependent activity toward DPP-IV enzyme, as reported by Tapadia et al. [25] and from our results. Moreover, these enzymatic assays are based on purified enzymes to evaluate their bioactivity and do not consider the several factors that can influence peptide bioactivity. It would be necessary to use methods that take into account intestinal environmental parameters that could have an effect on enzymes’ activity. Indeed, Lammi et al. [41] highlighted two peptides from soy and lupin proteins that showed inhibitory activity toward DPP-IV enzyme. With in vitro experiments, they were able to record reductions of 46% and 35% from soy and lupin peptides, respectively. However, in 2018, they carried out the same inhibitory study on Caco-2 cells and ex vivo human serum. In this more complex environment, the inhibitory activity recorded was 18.1% and 24.7% from lupin and soy, respectively, at the same concentration of the previous study of 100 µM of peptides [49].

Regulating the glucose transporters or insulin production expression can be an effective option for the attainment of a proper glycaemic control. Our results show a minimal increase in glucose transporters expression. The literature reports that some legume peptides are able to decrease the expression of SGLT1 and GLUT2 glucose transporters affecting the protein expression pathway and decreasing the translocation to the membrane [28]. After 24 h of treatment with peptides from soy hydrolysates, Lammi et al. [50] highlighted an increase in GLUT4 protein in hepatic cells. The same was documented by Munoz et al. [43] after 4 h of treatment, as fluorescence intensity quantified in the membrane region of the adipocytes to evaluate GLUT4 membrane translocation. Sandoval-Muniz et al. [18] studied different relevant gene expression including GLUT2 to understand the role of γ-C on glucose metabolism in diabetic rats. The authors showed that the protein has a positive effect but some results can be explained by other compounds’ effects or protein half-life modulation.

Some studies have been carried out on real food matrices enriched with γ-C or whole lupin, making the outcome difficult to interpret due to several factors that can affect the results. In fact, in pasta experiment, a lower glycaemic level was measured but this result could have been affected by the lower carbohydrate content in the meal if compared with standard pasta [16]. Further, in Dove et al.’s [22] study with legume-based beverages, the glucose-reducing effects may be ascribable to the fibre content, and the increase in viscosity and gastrointestinal solids, rather than a nutraceutical effect of legume proteins. In a white bread study, the authors suggest that the lower glycaemic index may have been influenced by a higher protein content; phytochemicals as oligosaccharides; phytic acid, tannins, and saponins content; or dietary fibre [24]. Other studies have observed differences in glucose and insulin responses in bread supplemented with lupin kernel flour, suggesting that the reduction in carbohydrate load is likely to be the primary contributor to the observed effects [51,52]. A study on lupin-enriched biscuits as a mid-meal snack for patients with type 2 diabetes was carried out by Skalkos et al. [53] and the favourable effect on glycaemia can be due to low GI, high protein, high fibre, and prebiotic content of lupin biscuits. Ward et al. [54] studied the effect of foods enriched with lupin on blood glucose level and concluded that there is no significant effect on glycaemic control, circulating fasting glucose, or insulin level. These studies suggest that the hypoglycaemic activity seems to depend on other components of food such as high protein content, dietary fibre, or phytochemicals content and is not related to γ-C.

Overall, and as a matter of fact, only limited activity has been evidenced for γ-C, which could be related to other factors independent to the activity of the protein itself. For example, in the case of pasta supplemented with a γ-C extract, the hypoglycaemic effect described could be related to the lower carbohydrate content rather than a direct contribution mediated by γ-C [16]. Several studies have been performed on dyslipidaemias in humans [23,55,56], in vitro [42], and in vivo [57,58,59]. Moreover, in this case, little evidence was collected on the direct effect of γ-C on lipid metabolism, leaving open the question if what was observed could be attributable to other molecules present in the preparations, such as the presence of Arginine and Lysine [56,58]. Finally, anti-inflammatory properties were evaluated both in vitro [41,45,60,61,62,63] and in vivo [64]; significant results on the reduction of the inflammation state were obtained only at extremely high doses of 4000 mg/kg [65]; furthermore, a comparative control with inert proteins was not performed.

## 5. Conclusions

In conclusion, available experimental evidence seems to indicate that lupin may provide some useful health benefits when added to diet but these effects are not likely to be directly ascribable to γ-C. Additionally, Bryant et al. [66] highlighted divergent results among studies and suggested it is the whole lupin that provides a broader range of health benefits.

## Figures and Tables

**Figure 1 nutrients-14-03666-f001:**
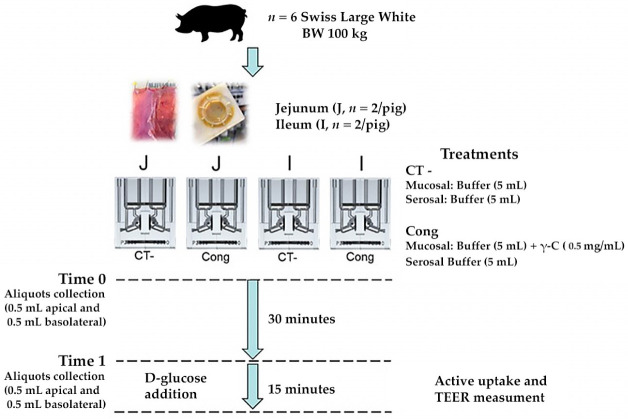
Study design of Ussing chamber experiments by using jejunum and ileum segments obtained by finishing pigs.

**Figure 2 nutrients-14-03666-f002:**
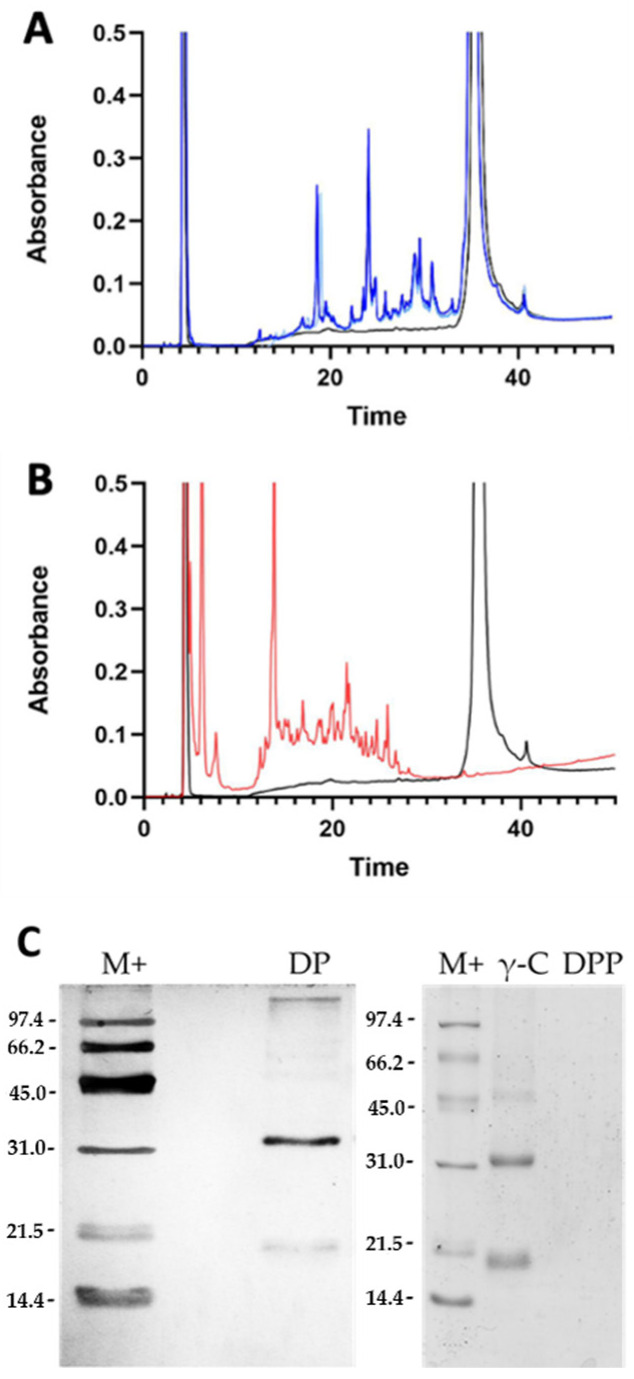
Analysis of the peptides obtained from γ-conglutin (γ-C) in vitro digestion. (**A**) RP-HPLC profiles of digestion of γ-C with trypsin only. The black line is the undigested γ-C, whereas the light and dark blue lines indicate the products of trypsin proteolysis (DP) after 15 and 45 min of digestion, respectively. (**B**) RP-HPLC of the γ-C double digestion with pepsin and pancreatin (45 min). The black line indicates the undigested γ-C, whereas the red line relates to the digestion products with pepsin and pancreatin enzymes (DPP). (**C**) SDS-PAGE under reducing conditions of intact protein (γ-C) and digested samples. DP: trypsin digested γ-C; DPP: pepsin and pancreatin digested γ-C.

**Figure 3 nutrients-14-03666-f003:**
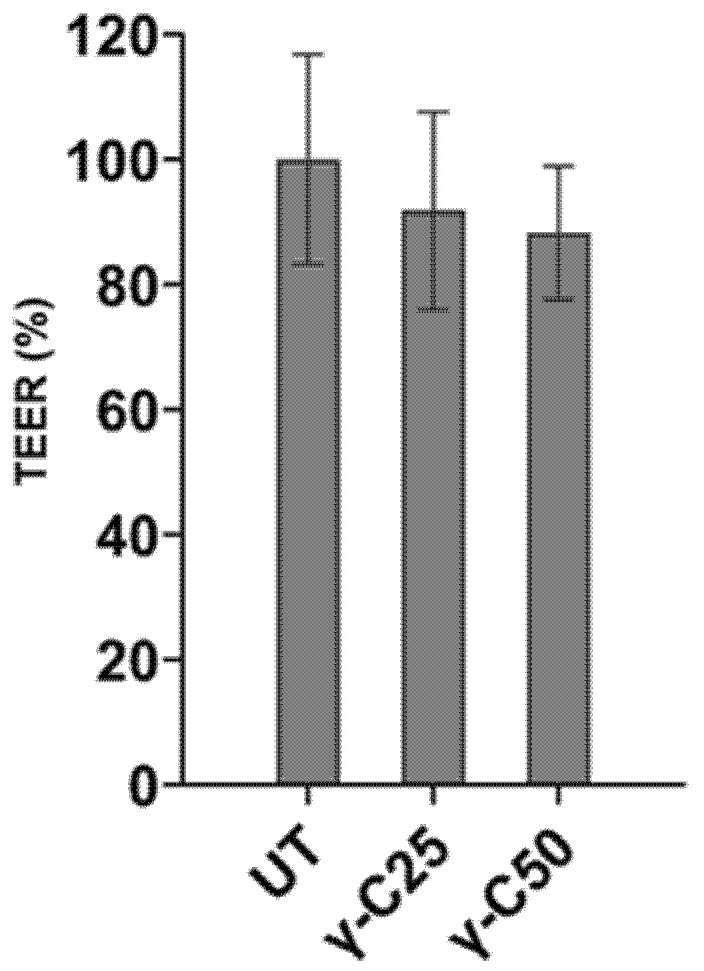
Transepithelial resistance (TEER) measured on IPEC-J2 cell line incubated for 24 h with 0.25 mg/mL (*v*/*v*, γ-C/culture medium, γ-C25) and 0.50 mg/mL (*v*/*v*, γ-C/culture medium, γ-C50 group) compared with the untreated group (UT). Data are expressed as mean ± standard deviation (*p* > 0.05).

**Figure 4 nutrients-14-03666-f004:**
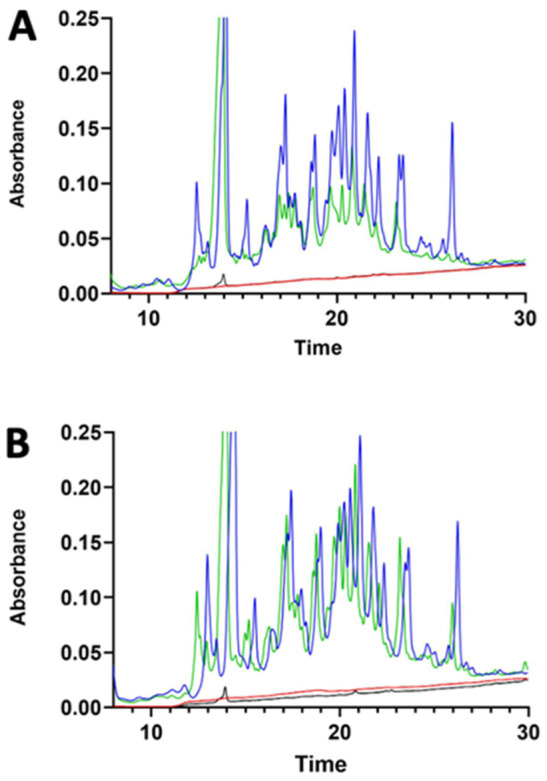
RP-HPLC chromatograms of the jejunum (**A**) and ileum (**B**) apical and basolateral sides of Ussing chamber at the beginning of the experiment (blue and red lines, respectively) and after 30 min of incubation with γ-C50 sample (green and black lines, respectively). γ-C50: 0.50 mg/mL of intact γ-C.

**Figure 5 nutrients-14-03666-f005:**
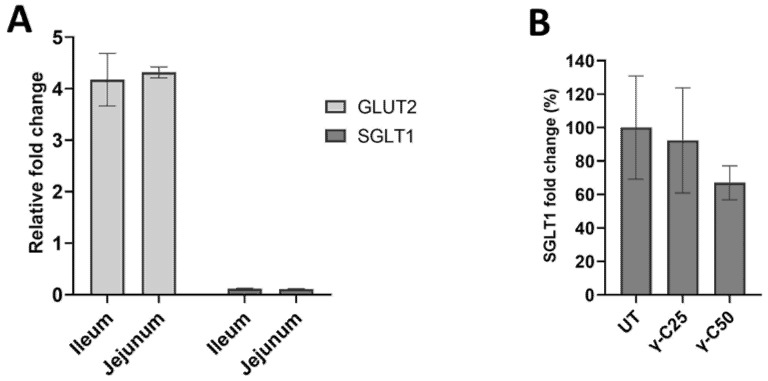
Expression of SGLT1 and GLUT2 glucose transporters in intestinal tissues and the IPEC-J2 cell line. (**A**) Expression fold determined by RT-qPCR of SGLT1 and GLUT2 (*p* < 0.05). (**B**) Effects of the 24 h incubation of IPEC-J2 cells with 0.25 mg/mL (*v*/*v*, γ-C/culture medium, γ-C25) and 0.50 mg/mL (*v*/*v*, γ-C/culture medium, γ-C50 group) on SGLT1 protein expression, compared with the untreated group (UT) (*p* > 0.05). Data are expressed as mean ± standard deviation.

**Figure 6 nutrients-14-03666-f006:**
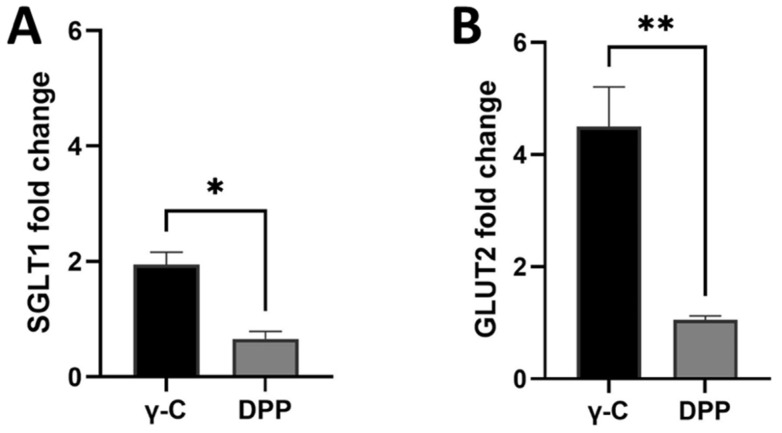
Effects of the 4-h incubation of undifferentiated Caco-2 cells with 1.00 mg/mL (γ-C) and pepsin and pancreatin digested sample (DPP) on SGLT1 (**A**) and GLUT2 (**B**) gene expression. Data are expressed as mean ± standard deviation. * *p* ≤ 0.05, ** *p* ≤ 0.01.

**Figure 7 nutrients-14-03666-f007:**
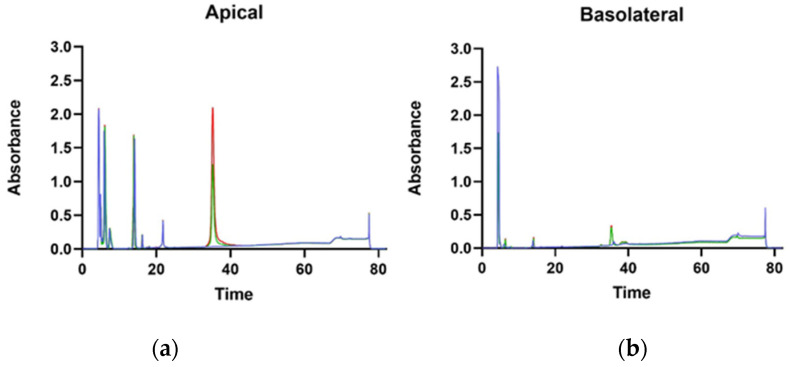
HPLC of supernatant from apical (**a**) and basolateral (**b**) chambers of transwell insert. Blue lines, red lines, and green lines indicate untreated, γ-C100 (1.00 mg/mL of γ-C), and γ-C50 (0.50 mg/mL of γ-C) samples, respectively.

**Figure 8 nutrients-14-03666-f008:**
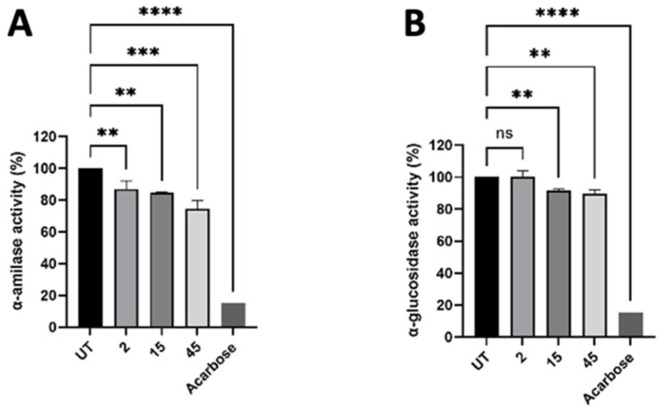
Effect of γ-C-derived peptides at 2, 15, and 45 min of digestion on α-amylase (**A**) and α-glucosidase (**B**) enzymes’ activity. Values are mean ± standard deviation of three independent experiments. **** *p* ≤ 0.0001, *** *p* ≤ 0.001, ** *p* ≤ 0.01. ns states for not significative.

**Table 1 nutrients-14-03666-t001:** D-glucose-induced ΔIsc (μA) and transepithelial resistance (TEER) measured on IPEC-J2 cell line incubated 24 h with 0.25 mg/mL (*v*/*v*, γ-C/culture medium, γ-C25) and 0.50 mg/mL (*v*/*v*, γ-C/culture medium, γ-C50). Results obtained in vitro were confirmed by the ex vivo experiments and compared with the untreated group (UT). Forskolin-induced ΔIsc (ΔIsc-Forsk) was evaluated to test cell viability at the end of the experiment. Data are expressed as mean ± standard errors of the means (*n* = 3 independent experiments).

	UT	γ-C25	γ-C50	SEM	*p*-Values
ΔIsc-Glucose (µA)	0.18	0.08	0.06	0.02	0.02
Δlsc-Forsk (µA)	1.07	0.98	0.95	0.05	0.64
TEER (Ω)	100	91.8	88.2	3.35	0.36

**Table 2 nutrients-14-03666-t002:** D-glucose-induced ΔIsc (μA) and transepithelial resistance (TEER) were measured on pig jejunum and ileum incubated for 30 min with 1 mg/mL (*v*/*v*, γ-C/culture medium) compared with the untreated group (UT). Data are expressed as means with their standard errors (*n* = 6 independent experiments). Abbreviations: T = Treatment; S = Segment.

	UT		γ-C			*p*-Values		
	Jejunum	Ileum	Jejunum	Ileum	SEM	T	S	TxS
Δ-Isc (µA)	6.77	9.18	3.57	7.89	1.07	0.02	0.27	0.64
TEER (Ω)	38.3	58.9	42.7	78.4	7.62	0.11	0.08	0.96

## Data Availability

Not applicable.

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
