# Peer review of "Lupinus albus γ-Conglutin: New Findings about Its Action at the Intestinal Barrier and a Critical Analysis of the State of the Art on Its Postprandial Glycaemic Regulating Activity"

_nutrients, 2022, doi:10.3390/nu14173666_

Round 1

Reviewer 1 Report

The presented study provide very interesting data about possible mechanism of action of lupin seed hypoglycaemic protein – gamma-conglutin. The authors used in vitro as well as ex vivo systems and undertook attempts to show detailed mechanism of action/absorption utilizing enzymatic assess and protein expression determinations. The obtained results however, did not provided clear explanation for the unique phenomenon of gamma-conglutin. Nevertheless, the manuscript provide next step towards understanding the mechanism of action. Noteworthy, the experimental design as well as description of the methods was provided with high accuracy. Therefore, this study require only minor correction prior future consideration in the Journal

L 12: There are a number of other glycoproteins in Lupinus albus seed, thus, the term „major” is appropriate.

Figure 2. What is possible explanation that on RP-HPLC chromatograms (a) there is peak of gamma-conglutin (DP sample) but on SDS-PAGE gel there is no bands?

Author Response

Reviewer 1

We thank the reviewer for the positive comments.

Question: L 12: There are a number of other glycoproteins in Lupinus albus seed, thus, the term „major” is appropriate.

 Answer: We agree with the reviewer the “major” has been removed.

Question: Figure 2. What is possible explanation that on RP-HPLC chromatograms (a) there is peak of gamma-conglutin (DP sample) but on SDS-PAGE gel there is no bands?

 Answer: We agree with the reviewer. The picture was not very clear, we have now changed the photographs in order to make evident the relevant polypeptides. The legend has been rephrased accordingly.

Reviewer 2 Report

This manuscrips describes the interactions of  γ-conglutin with porcine intestinal barrier, the contact point  of a dietary protein with the organism, and on  γ-conglutin influences on glucose uptake. The novelty of the design and results were slightly inadequate, moreover some flaws of the manuscript need to be addressed and improved.

The following are questions and suggestions:

1.This study does not provide ethics and permission for animal experimentation.

2.The purpose and significance of this study need to be specified.

3.The discussion section is too verbose and needs to be carefully condensed.

4.Description of picture 2C and related results is confused and unclear. Please describe in detail.

5.The contents and instructions presented in Picture 5 were unreasonable and need to be corrected carefully.

6.The serial number of the result section is incorrectly marked, there are two “3.2”.

7.The grammar and writing of the manuscrips need to be revised carefully. 

Q1: Please revise line 98 “both intact and digested, in the apical chamber”. This sentence is misleading. Should commas be removed?

Q2: In section “2.4. SGLT1 and GLUT2 glucose transporters gene expression”, why Caco-2 cells and pig intestines use the same reagents but different q-PCR conditions?

Q3: In section 3.2, please state "TEER of the IPEC-J2 cells monolayer compared to the control" P-value and indicated in the figure legend.

Q4: What does Figure 5 want to illustrate? I did not find the relevant description of Figure 5. And the quality of Figure 5 is not satisfactory, there is no marker in the untreated group.

Q5: Figure 6. Please indicate the gene represented by color in the figure, not in the legend. Please indicate the P value.

Q6: The serial number of the result section is incorrectly marked, there are two “3.2”.

Q7: Please mark the P value in the description and legend section of Figure 7.

Q8: Please condense the discussion section and focus on the main points of the article.

Q9: Please revise details. For example, in materials and methods part, please modify “2.2α-. Glucosidase and α-amylase inhibition assay”; In line 80 and 87, please modify “))”; In line 95, the first letter after the period should be capitalized; Please modify “3.1.γ-. conglutin hydrolysate characterization” and so on.

Q10: There are many capitalization problems in the article, please check carefully.

Q11: For parts “not shown”, it is hoped that this can be expressed in the manuscript, or added in the supplementary file.

Q12: Hope to present each part of the pictures as a group picture.

Q13: Please mark each abbreviation at the end of the figure legend.

Q14: The full name of the abbreviation is marked when it first appears. There are a lot of mixed use of full name and abbreviation in the text. Please modify.

Q15: Please unify the number of reserved digits after the decimal point in the full text.

Author Response

Reviewer 2

Question: This study does not provide ethics and permission for animal experimentation.

Answer: The ethical statement has been provided in the Institutional Review Board Statement section at the end of the test and a pertinent sentence is also reported in M&M

Question: The purpose and significance of this study need to be specified.

Answer: We agree with the reviewer the significance and purpose are “fuzzy”. We modify the text (now at rows 32-34) in order to make the purposes clear.

Question: The discussion section is too verbose and needs to be carefully condensed.

Answer: The discussion section has been revised in order to make it less dispersive and more focused. The redundant parts have been removed. We take the liberty to point out that the discussion of this article is also intended as a critical review of the literature, to mitigate and to make a boundary to the tendency to study the effects of dietary proteins on cells/systems other than intestinal ones which likely never came in contact with dietary proteins.

Question: Description of picture 2C and related results is confused and unclear. Please describe in detail.

Answer: The figure, also taking into account a comment from another reviewer, has been slightly changed. Thus the legend has been rephrased and made clearer. In addition, more details about the Figure 2C have been added to the results section (line 242-244).

Question: The contents and instructions presented in Picture 5 were unreasonable and need to be corrected carefully.

Answer: We agree with the reviewer; Figure 5 is not strictly relevant. The figure is now provided as supplementary file S2 including the spectrometry identification of albumin. See also answer to Q4.

Question: The grammar and writing of the manuscrips need to be revised carefully. 

Answer: Typos and errors have been corrected and the text has been revised by a native English.

Q1: Please revise line 98 “both intact and digested, in the apical chamber”. This sentence is misleading. Should commas be removed?

Answer: We agree with the reviewer; the comma has been removed.

Q2: In section “2.4. SGLT1 and GLUT2 glucose transporters gene expression”, why Caco-2 cells and pig intestines use the same reagents but different q-PCR conditions?

Answer: The experimental conditions are different because Caco-2 cells are human while the intestine is from pig. The primers selected from the literature [28,29] have different sequences that require different PCR conditions.

Q3: In section 3.2, please state "TEER of the IPEC-J2 cells monolayer compared to the control" P-value and indicated in the figure legend.

Answer: The detail has been added to the text, line 265.

Q4: What does Figure 5 want to illustrate? I did not find the relevant description of Figure 5. And the quality of Figure 5 is not satisfactory, there is no marker in the untreated group.

Answer: The figure is now in supplementary file as answered before, with implemented description and marker added.

Q5: Figure 6. Please indicate the gene represented by color in the figure, not in the legend. Please indicate the P value.

Answer: The figure has been corrected and P values indicated.

Q6: The serial number of the result section is incorrectly marked, there are two “3.2”.

Answer: The reviewer is right; the numbering has been corrected.

Q7: Please mark the P value in the description and legend section of Figure 7.

Answer: The figure now reports the p-value among the samples.

Q8: Please condense the discussion section and focus on the main points of the article.

Answer: The discussion section has been revised in order to make it less dispersive and more focused. The redundant parts have been removed. We take the liberty to point out that the discussion of this article is also intended as a critical review of the literature, to mitigate and to make a boundary to the tendency to study the effects of dietary proteins on cells/systems other than intestinal ones which likely never came in contact with dietary proteins.

Q9: Please revise details. For example, in materials and methods part, please modify “2.2α-. Glucosidase and α-amylase inhibition assay”; In line 80 and 87, please modify “))”; In line 95, the first letter after the period should be capitalized; Please modify “3.1.γ-. conglutin hydrolysate characterization” and so on.

Answer: The specified details have been carefully revised according to reviewer suggestion. All the text has been checked for gross errors and typos

Q10: There are many capitalization problems in the article, please check carefully.

Answer: The paper has been revised for capital letters mistakes

Q11: For parts “not shown”, it is hoped that this can be expressed in the manuscript, or added in the supplementary file.

Answer: Data previously indicated as not shown are now provided as supplementary material (S1-S5)

Q12: Hope to present each part of the pictures as a group picture.

Answer: We don’t understand the comment. We have checked carefully all the pictures in order to solve any possible incongruity.

Q13: Please mark each abbreviation at the end of the figure legend.

Answer: All abbreviation have been explained in figure legends

Q14: The full name of the abbreviation is marked when it first appears. There are a lot of mixed use of full name and abbreviation in the text. Please modify.

Answer: All abbreviations are now indicated when they first appear.

Q15: Please unify the number of reserved digits after the decimal point in the full text.

Answer: The numbers have been checked and corrected.